# The prevalence, incidence, and risk factors of mental health problems and mental health service use before and 9 months after the COVID-19 outbreak among the general Dutch population. A 3-wave prospective study

**Peter G. van der Velden** [1,2]*, **Miquelle Marchand**[1], **Marcel Das**[1,3], **Ruud Muffels**[2], **Mark Bosmans**[4]

**1** Centerdata, Tilburg, The Netherlands, **2** Tranzo (Scientific Center for Care and Welfare), Tilburg School of Social and Behavioral Sciences, Tilburg University, Tilburg, The Netherlands, **3** Tilburg School of Economics and Management, Tilburg University, Tilburg, The Netherlands, **4** NIVEL, Utrecht, The Netherlands

* pg.vandervelden@tilburguniversity.edu, pg.vandervelden@centerdata.nl

**Data Availability Statement:** The study was conducted using the Dutch Longitudinal Internet

## Abstract

### Objectives

Gain insight into the effects of the COVID-19 pandemic on the prevalence, incidence, and risk factors of mental health problems among the Dutch general population and different age groups in November-December 2020, compared with the prevalence, incidence, and risk factors in the same period in 2018 and 2019. More specifically, the prevalence, incidence, and risk factors of anxiety and depression symptoms, sleep problems, fatigue, impaired functioning due to health problems, and use of medicines for sleep problems, medicines for anxiety and depression, and mental health service.

### Methods

We extracted data from the Longitudinal Internet studies for the Social Sciences (LISS) panel that is based on a probability sample of the Dutch population of 16 years and older by Statistics Netherlands. We focused on three waves of the longitudinal Health module in November-December 2018 (T1), November-December 2019 (T2), and November-December 2020 (T3), and selected respondents who were 18 years and older at T1. In total, 4,064 respondents participated in all three surveys. Data were weighted using 16 demographics profiles of the Dutch adult population. The course of mental health problems was examined using generalized estimating equations (GEE) for longitudinal ordinal data and differences in incidence with logistic regression analyses. In both types of analyses, we controlled for sex, age, marital status, employment status, education level, and physical disease.

### Results

Among the total study sample, no significant increase in the prevalence of anxiety and depression symptoms, sleep problems, fatigue, impaired functioning due to health

studies for the Social Sciences (LISS) pane. The LISS panel started in 2007 and is based on a large traditional probability sample drawn from the Dutch population. The Dutch Research Counsel (NWO) funded the set-up of LISS. Panel members receive an incentive of €15 per hour for their participation and those who do not have a computer and/or Internet access are provided with the necessary equipment at home. Further information about all conducted surveys and regulations for free access to the data can be found at https://www.dataarchive.lissdata.nl/ (in English). The LISS panel has received the international Data Seal of Approval (see https://www. datasealofapproval.org/en/). All data of studies conducted with the LISS panel are anonymized.

**Funding:** The author(s) received no specific funding for this work.

**Competing interests:** The authors have declared that no competing interests exist.

problems, use of medicines for sleep problems, of medicines for anxiety and depression, and of mental health service in November-December 2020 was observed, compared with the prevalence in November-December 2018 and 2019 (T3 did not differ from T1 and T2). Among the four different age categories (18–34, 35–49, 50–64, and 65 years old and older respondents), 50–64 years respondents had a significantly lower prevalence of anxiety and depression symptoms at T3 than at T1 and T2, while the prevalence at T1 and T2 did not differ. A similar pattern among 65+ respondents was found for mental health service use. We found no indications that the incidence of examined health problems at T2 (no problems at T1, problems at T2) and T3 (no problems at T2, problems at T3) differed. Risk factors for mental health problems at T2 were mostly similar to risk factors at T3; sex and age were less/not a risk factor for sleep problems at T3 compared with at T2.

## Conclusions

The prevalence, incidence, and risk factors of the examined mental health problems examined nine months after the COVID-19 outbreak appear to be very stable across the end of 2018, 2019, and 2020 among the Dutch adult population and different age categories, suggesting that the Dutch adult population in general is rather resilient given all disruptions due to this pandemic.

## Introduction

The ongoing COVID-pandemic and preventive measures to contain the pandemic as much as possible have profound negative effects on affected countries and their residents. These effects vary from, but are not restricted to, higher death rates, recovery problems among infected, overloaded hospitals, closed schools, and diminished social contact, job loss, repeated lockdowns, political tensions, diminished economic growth, and large governmental financial debt. The global weekly Operational Update on COVID-19 of the World Health Organization of November 6 2020 and December 21 2020 [1], the period in which the last survey of the present study was conducted, reported 1,231,017 and 1,690061 deaths respectively and 48,534,508 and 75,704,857 confirmed COVID-19 cases respectively.

Although effective vaccines became available in 2021 [2], the question to what extent this pandemic affected the mental health of the general population is and remains important. Based on the Conservation of Resources (COR) theory of Hobfoll [3, 4] we may expect that the pandemic has negative effects among the general population since it directly or indirectly threatens important resources such as safety and health (for instance by being infected, loss of a significant other), social contacts and support (for instance by social distancing and staying-at-home following lockdowns), work and income (for instance by loss of job or reduced work) among the general population. Resource loss is an important factor, like existing problems, in predicting the impact of stressful events on mental health such as this pandemic. However, according to the COR [3, 4] people also strive to obtain, retain, protect resources, and to restore lost resources (such as social contacts, employment, housing, health) and their resilience should not be underestimated [3–6]. The duration of this pandemic and effects may nevertheless undermine the capacity of individuals, communities, and countries to cope with the negative effects of this pandemic on the medium and longer term. This may cause stress and increase the risk for mental health problems.

To gain insight into the effects on mental health, prospective studies based on probability samples of the general population with pre-COVID-19 data are warranted. However, compared with the large number of cross-sectional COVID-19 studies that are often based on convenience samples for which the representativeness is unclear [7, 8], the number of prospective studies based on probability samples is relatively limited. In the beginning of 2021, we identified 12 prospective peer-reviewed studies that were based on probability samples of the general population with nonretrospective data on pre-COVID-19 mental health. Below we first provide a brief summary of the main outcomes of these studies. Given the aim of the present study, we focus on the prevalence of mental health problems among the total study samples and different age categories.

With respect to the UK, the study by Pierce et al. [9] showed that mean scores on the General Health Questionnaire (GHQ-12) increased significantly from 11.5 in 2018–2019 (data were collected year-round) to 12.6 in April 2020. This increase was not considered a simple continuation of previous upwards trends from 2014 to 2019. The prevalence of clinically significant mental distress increased from 18.9% in 2018–2019 to 27.3% in April 2020. The increase in mental distress appeared to be the largest in respondents 18–34 years old. The study by Proto and Quintana-Domeque [10] showed similar findings up to April 2020 but focused especially on ethnicity and gender. Niedzwiedz et al. [11] found that the proportion of people drinking four or more times per week increased (Relative Risk (RR) = 1.4) as did binge drinking (RR = 1.5). Daly et al. [12] analyzed data on mental health up to June 2020. Their results showed that the prevalence of mental health problems increased from 24.3% in 2017–2019 to 37.8% in April, 34.7% in May, and 31.9% in June 2020. Although elevated in June 2020 compared with 2017–2019, the prevalence was lower (-5.9%) than in April 2020. Respondents of 18–34 years old showed the largest increase in the prevalence of mental health problems.

Of the prospective studies in the USA, Twenge and Joiner [13] compared anxiety and depression symptomatology among adults in the *National Health Interview Survey* (NHIS; January-June 2019) and the *Household Pulse Survey* (HPS; April-May 2020). Respondents in the HPS were three to four times more likely to screen positive for anxiety or depression disorders compared with respondents in the NHIS. McGinty et al. [14] using data of the NHIS (2018) and the *Johns Hopkins COVID-19 Civic Life and Public Health Survey* (April 2020), found that the prevalence of psychological distress increased from 3.9% to 13.6%, and the prevalence increased the most among young adults aged 18 to 29 years (4% to 24.0%). Daly et al. [15] used data of the *National Health and Nutrition Examination Survey* (NHNE, 2017–2018) and the *Understanding America Study* (UAS, March and April 2020). Results showed a significant higher prevalence of depression symptoms in March (10.6%) and April 2020 (14.4%) than before the outbreak (8.7%). Among 18–34 years old respondents, the increase was 7.3% between pre-COVID and March 2020, and 13.4% between March and April 2020. Among older respondents, a smaller increase between March and April 2020 was found. Breslau et al. [16] found that the prevalence of clinical psychological distress in February 2019 (10.9%) was as high as in May 2020 (10.2%). Among 20–39 and 40–59 years old respondents, the prevalence of respondents who suffered from an increase in problems (20.8% and 14.4% respectively) was higher than among those of 60 years and older. Ettman et al. [17] compared the prevalence of depression symptoms among respondents of the *COVID-19 Life Stressors Impact on Mental Health and Well-being study* conducted in the period March 31, 2020-April 13, 2020 and, like Daly et al. [15], and respondents of the NHNE Survey conducted in 2017–2018. Results showed that the prevalence of mild and severe symptoms was higher during COVID-19 than before (mild: 24.6% vs 16.2%, severe: 5.1% vs 0.7%). In contrast to pre-COVID-19, during the pandemic moderate to severe depression symptom levels differed between age categories with a higher prevalence among younger adults.

With respect to the Netherlands, Van der Velden et al. [18] found, like Breslau [16], no increase in the prevalence of anxiety and depression symptoms among adults in March 2020 (17.0%) compared with the prevalence in November 2019 (16.9%). Compared with 18–34 years old respondents (19.7%), 35–49 years respondents had a significant higher prevalence (22.1%) and 65 years and older respondents had a lower prevalence of symptoms (10.6%) in March 2020. The study by Van Tilburg et al. [19] focused on respondents of 65 years and older. Results showed a significant but trivial improvement in May 2020 compared with November 2019. The follow-study by Van der Velden et al. [20] with data up to June 2020, showed a significant but small decrease in the prevalence of anxiety and depression symptoms in June (15.3%) compared with March 2020 (17.2%) and November 2019 (16.8%). In addition, they found that the recovery of symptoms in the period November 2019- March 2020, did not differ significantly from the period March 2020 to June 2020.

The aforementioned studies mainly focused on psychological distress, anxiety, and depression symptoms, and did not provide insight into the effects of the COVID-19 pandemic on the *prevalence* of other relevant mental health problems such as fatigue and sleep problems, and other indicators of mental health such as the use of medicines for anxiety and depression symptoms, use of medicines for sleep problems, mental health service use, and impaired functioning due to health problems. Moreover, little is known about the post-COVID-outbreak *incidence* of mental health problems compared with the incidence of mental health problems before the COVID-19 outbreak. Finally, very little is known about the extent to which prospective risk factors of mental health problems *before* the outbreak are also prospective risk factors of problems and service use *after* the outbreak [18].

With respect to COVID-19 in the Netherlands, soon after the outbreak, the Dutch government implemented large financial support programs for companies who significantly lost revenues because of the COVID-19 pandemic to allow them to keep people employed. In addition, governmental taxes were postponed. The Dutch governmental deficit increased in 2020 with 40 billion euro to 435 billion euro largely due to these financial programs [21]. In 2020, about 800,000 residents were tested positive for COVID-19, although the number of residents with COVID-19 in 2020 is presumably higher [22]. After the outbreak, residents and companies were confronted with several (partial) lockdowns in 2020, including closed schools and universities. Statistics Netherlands (CBS) reported that about 169,000 had died in the Netherlands, 10% more than expected compared with previous years [23].

The aim of the present prospective population-based study is to fill the above identified gaps of scientific knowledge. For this purpose, data on health were extracted from surveys conducted with the LISS panel in November-December 2018 (T1), November-December 2019 (T2), and November-December 2020 (T3), nine months after the COVID-19 outbreak. It is part of an ongoing COVID-19 study [18, 20, 24] using this longitudinal panel. Research questions of the present study were:

1. To what extent did the *prevalence* of mental health problems and mental health service use among the general population before the COVID-19 outbreak in November-December 2018 (T1) and 2019 (T2), differ from the prevalence after the COVID-19 outbreak in November-December 2020 (T3)?

2. To what extent did the *incidence* of mental health problems and mental health service use among the general population before the COVID-19 outbreak in November-December 2019 (T2), differ from the incidence after the COVID-19 outbreak in November-December 2020 (T3)?

3. To what extent were well-documented *risk factors of* mental health problems and mental health service use before the outbreak (November-December 2019, T2), comparable with risk factors of problems and service use after the outbreak (November-December 2020, T3), such as sex, age, education level, marital status, employment status and physical health [25–28].

## Materials and methods

### Procedures and participants

For the present study, data on mental health were extracted from the Longitudinal Internet studies for the Social Sciences (LISS) panel [29]. This panel is based on a traditional probability sample drawn from the Dutch population register of 16 years and older by Statistics Netherlands and administered by Centerdata. People cannot sign up themselves as respondents for the LISS panel. The set-up was funded by the Dutch Research Council (NWO). Panel members receive an incentive of 15 euros per hour and members who do not have a computer and/or internet access are provided with the necessary equipment at home (for further information about the LISS panel, all conducted studies since 2007, and open access data see: https://www.dataarchive.lissdata.nl; in English).

Data on mental health of adults were extracted from the longitudinal Health module, in particular the waves in November 2018 (T1: $N^{invited}$ = 6,466, response = 84.4%), in November 2019 (T2: $N^{invited =}$ 5,954, response = 86.4%), and in November 2020 (T3: $N^{invited}$ = 6,832, response = 83.6%), with reminders in December of each year. In total, 4,107 respondents who were 18 years or older at T1, participated at T1, T2, and T3. The total study sample consisted of 4,064 respondents with complete data across the three surveys (99%). We next weighted the data using 16 exclusive demographic profiles among the total adult Dutch population to optimize the representativeness of the current study, based on the data of Statistics Netherlands (see: https://opendata.cbs.nl/#/CBS/en/; in English). The 16 profiles were constructed using the variables sex (male, female), age (18–34, 35–49, 50–64, 65 years and older), and marital status (married and unmarried), yielding $2*4*2$ = 16 demographic profiles. All findings are based on the weighted sample.

### Ethical approval and informed consent

Since our research did not impose certain (experimental) behavior, our research did not need the approval of a Dutch Medical Ethical Testing committee according to the Dutch Law (see https://english.ccmo.nl/investigators/legal-framework-for-medical-scientific-research/your-research-is-it-subject-to-the-wmo-or-not). Nevertheless, the Health module (as part of the Longitudinal Core Study in LISS, starting in 2007) was evaluated and approved by the Board of Overseers, an Internal Review Board (IRB) until 2014. In accordance with the General Data Protection Regulation (GDPR), participants gave explicit written consent for the use of the collected data for scientific and policy relevant research.

### Measures

We used the following six measures administered in each survey to obtain insight into the mental health problems and mental health service use of respondents in November-December 2018, 2019, and 2020. At each survey respondents' sex, age, marital status, and employed status (primary occupation) was assessed. For the present study, marital status and employed status were recoded into married (1 = yes, 2 = no) and employed (1 = yes, 2 = no).

**Anxiety and depression symptoms.** Anxiety and depression symptoms were examined using the Mental Health Index or Inventory (5-item subscale of the Medical Outcomes Study

(MOS), 36-Item Short Form Survey Instrument (SF-36, [30, 31])). Respondents were asked to rate their mental health during the past month on 6-point Likert scales (0 = never to 5 = continuously). After recoding the negatively formulated items (items 1, 2, and 4), the total scores were computed and multiplied by four (range 0 to 100, all Cronbach's $\alpha > .85$). Lower scores reflect higher symptom levels. A cut-off of $\leq 59$ was used to identify respondents with moderate to severe symptom levels and a cut-off of $\leq 44$ to identify respondents with severe symptom levels [32].

**Fatigue and sleep problems.** Respondents were furthermore administered a list of 10 (physical) problems people may suffer from, varying from heart complaints to sleeping problems. For the present study, we focused on the items '*Do you regularly suffer from fatigue'* and '*Do you regularly suffer from sleep problems*' (1 = no, 2 = yes).

**Impaired functioning due to (mental) health problems.** Psychopathology is consistently and independently associated with increased disability [31]. Health-related impaired functioning were assessed with the question '*To what extent did your physical health or emotional problems hinder your work over the past month, for instance in your job, the housekeeping, or in school*? This question is comparable with questions of the MOS-36 [33] and the European Health Interview Surveys (SILC-EU) [34], and had a 5-point Likert scale (1 = not at all to 5 = very much). For the present study, the scores were recoded into low (1 = 1,2,3) and high (2 = 4,5).

**Medicines for anxiety and depression.** Use of medicines were assessed for several conditions varying from medicines for blood pressure to sleep problems ('*Are you currently taking medicine at least once a week for .. '*). In the present study, we focused on the current use of medicines for anxiety and depression symptoms and sleep problems (0 = no, 1 = yes).

**Mental health service use.** Mental health service use (MHS) was assessed by one question '*How often did you use the following health services over the past 12 months*?', with answer categories varying from psychiatrist/ psychologist/ psychotherapist to dentist. For the present study, we focused on the use of a psychiatrist, psychologist, or psychotherapist and recoded 'use' into 'no' (0 = no use) and 'yes' (1 = once or more) in the past 12 months.

**Physical disease.** Respondents were administered a list of 19 physical diseases/ problems and asked if they had one or more of these diseases/problems according to a physician ('*Has a physician told you this last year that you suffer from one of the following diseases/ problems*?') varying from cancer to chronic lung disease. For the present study, we recoded the answers into 'no disease' (1 = none of the diseases/problems included in 'diseases') and 'disease' (2 = angina; pain in the chest a heart attack including infarction or coronary thrombosis or another heart problem including heart failure; stroke or brain infarction or a disease affecting the blood vessels in the brain; diabetes or a too high blood sugar level; chronic lung disease such as chronic bronchitis or emphysema; asthma; arthritis, including osteoarthritis, or rheumatism, bone decalcification or osteoporosis; cancer or malignant tumor, including leukemia or lymphoma; and/or benign tumor (skin tumor, polyps, angioma)).

## Statistical methods

To examine the extent to which the prevalence of the seven assessed mental health problems after the COVID-19 outbreak in November-December 2020 (T3) changed compared with the prevalence in November-December 2018 (T1) and 2019 (T2), generalized estimating equations (GEE) for longitudinal ordinal data were conducted (GENLIN in SPSS version 28, using an autoregressive working correlation structure) with problems at T1, T2 and T3 as dependent variables (separate analyses for each dependent variable). In the analyses, sex, age, marital status, employment status, education level, and disease at T1, T2, and T3 were used as control variables.

For the incidence at T3 (prevalence of new cases), the prevalence of mental health problems and use at T3 was assessed among those *without* these problems and or use at T2. Likewise, for the incidence at T2, problems and use at T2 were assessed among those *without* these problems and use at T1 (the incidence at T1 could not be computed among the current study sample). To assess the extent to which the incidence of mental health problems and use in November-December 2020 (T3) changed compared with the incidence in November-December 2019 (T2), multivariate logistic regression analyses were conducted as follows. The incidence at T2 and T3 cannot directly be compared because those with problems and use at T2 and T3 without problems and use at the previous survey partly overlap. To enable a comparison of the incidence at T2 and T3, we therefore first randomly split the total study sample into two almost equal independent subgroups of respondents (A: n = 2,025; and B: n = 2,019; because of the weighting the numbers of both groups slightly differs). In both subgroups, the incidence at T2 (A1, B1) and T3 (A2, B2) were computed. We finally compared the incidence of A1and B2, and compared the incidence of B1 and A2 using logistic regression with the same control variables (control variables at T1 for analyses incidence at T2, control variables at T2 for analyses incidence at T3).

Prospective risk factors of mental health problems and service use were examined using multivariate logistic regression analyses with problems and service use at T2 and T3 as dependent variables. The variables sex, age, education level, marital status, employment status, and physical disease at T1 and T2, respectively, were simultaneously entered as predictors (for instance, variables at T2 were entered as predictors for problems at T3).

## Results

### Nonresponse

Multivariate logistic regression analyses (before weighting) with nonresponse at T2 and T3 as the dependent variable (1 = participated at T1, T2 and T3, 2 = did not participate at T2 and T3) showed that the nonresponse was not significantly ($p > .05$) associated with the seven mental health and mental service use variables at T1, disease at T1, and not with sex, employment status, and education level at T1. Unmarried respondents (85.1%) compared with married respondents (90.4%) participated less at T1, T2, and T3 (adjusted Odds ratio (aOR) = 0.80, 95% confidence interval (95% CI) = 0.66–0.98, p = .029)). Compared with 18–34 years old respondents (79.1%), 34–50 years old (85.7%, aOR = 1.43, 95% CI = 1.10–1.87, p = .008), 50–64 years old (92.3%, aOR = 2.80, 95% CI = 2.10–3.75; p < .001), and 65 years or older respondents (90.7%, aOR = 2.10, 95% CI = 1.53–2.84, p < .001) participated significantly more often. As described, the study sample was weighted for sex, marital status, and age.

### Characteristics of respondents

The characteristics of the study sample are presented in Table 1. It shows, among others, that in absolute numbers as time passes by, more respondents are married, become older, have a higher education level, as well as more often have a physical disease. At T3, the youngest respondents were 20 years old.

### Prevalence of mental health problems and service use of total study sample

The prevalence of the assessed mental health problems and service use among the total study sample at T1, T2, and T3 are presented in Table 2. The prevalence of mental health problems and service use at T3 did not significantly differ from T1 and T2 except for impaired functioning due to health problems. The prevalence of impaired functioning due to health problems

**Table 1. Characteristics of respondents (N = 4,064).**

| | Survey | | |
|---|---|---|---|
| | **2018 (T1)** | **2019 (T2)** | **2020 (T3)** |
| | **n (%)** | **n (%)** | **n (%)** |
| Employed | | | |
| • no | 1854 (45.6) | 1825 (44.9) | 1867 (45.9) |
| • yes | 2210 (54.4) | 2239 (55.1) | 2197 (54.1) |
| Married | | | |
| • yes | 1957 (48.2) | 1972 (48.5) | 1999 (49.2) |
| • no | 2107 (51.8) | 2092 (51.5) | 2065 (50.8) |
| Education level[1] | | | |
| • low | 1013 (24.9) | 992 (24.4) | 969 (23.8) |
| • medium | 1466 (36.1) | 1441 (35.5) | 1416 (34.8) |
| • high | 1585 (39.0) | 1631 (40.1) | 1679 (41.3) |
| Sex | | | |
| • male | 2002 (49.3) | idem | idem |
| • female | 2062 (50.7) | idem | idem |
| Physical disease | | | |
| • no | 3303 (81.3) | 3266 (80.4) | 3253 (80.0) |
| • yes | 761 (18.7) | 798 (19.6) | 811 (20.0) |
| Age category | | | |
| • 18–34 years old | 1083 (26.6) | 1018 (25.0) | 933 (23.0) |
| • 35–49 years old | 961 (23.6) | 933 (23.0) | 941 (23.2) |
| • 50–64 years old | 1052 (25.9) | 1061 (26.1) | 1054 (25.9) |
| • 65 years old or older | 968 (23.8) | 1052 (25.9) | 1136 (28.0) |

[1] Low = primary school, intermediate secondary education, US: junior high school; Medium = higher secondary education/preparatory university education, US: senior high school, intermediate vocational education, US: junior college; High = higher vocational education, US: college, 6 university according to education level categories of Statistics Netherlands (CBS). All results are based on the weighted sample. Due to weighting, numbers may slightly differ between tables.

was significantly lower at T3 (8.5%) than at T1 (9.9%), but not significantly different from the prevalence at T2 (8.5%).

Analyses for anxiety and depression scores using the cut-off score of $\leq$ 44 of the MHI-5 showed similar outcomes: the prevalence of severe symptom levels at T1 (6.6%), T2 (6.5%) and T3 (6.3%) also did not differ significantly (results not shown in table).

## Prevalence of mental health problems and service use among age categories

Among respondents 18–34 years old, the prevalence of anxiety symptoms at T3 (22.3%) was significantly higher than at T1 ((19.9%, adjusted Odds Ratio (aOR) = 1.30, 95% Confidence interval (95% CI) = 1.08–1.57, p = .009)), but not compared with T2 (20.7%). In contrast, among respondents 50–64 years old, the prevalence of these symptoms at T3 (13.3%) was significantly lower than at T1 (14.6%, aOR = 0.82, 95% CI = 0.68–0.99, p = .043) and T2 (15.1%; aOR = 0.81, 95% CI = 0.68–0.95, p = .013).

With respect to sleep problems, among respondents 18–34 years old, the prevalence at T3 (14.6%) was significantly higher than at T1 (13.2%; aOR = 1.19, 95% CI = 1.02–1.39, p = .023), but not compared with T2 (14.8%).

**Table 2. Prevalence of mental health problems and service use (N = 4,064).**

| | November-December 2018 (T1) | November-December 2019 (T2) | November-December 2020 (T3) | T1 versus T3 | T2 versus T3 | T1 versus T2 |
|---|---|---|---|---|---|---|
| **Prevalence** | n (%) | n (%) | n (%) | aOR (95% CI) | aOR (95% CI) | aOR (95% CI) |
| Anxiety and depression symptoms[1] | 660 (16.2) | 687 (16.9) | 686 (16.9) | 1.09 (0.99–1.19) | 1.00 (0.92–1.09) | 1.08 (0.99–1.18) |
| Sleep problems | 834 (20.5) | 871 (21.4) | 869 (21.4) | 1.03 (0.98–1.09) | 0.99 (0.95–1.03) | 1.05 (1.00–1.09)* |
| Fatigue | 1275 (31.3) | 1292 (31.8) | 1266 (31.2) | 1.00 (0.96–1.05) | 0.98 (0.95–1.01) | 1.02 (0.99–1.06) |
| Impaired functioning due to health problems | 404 (9.9) | 345 (8.5) | 348 (8.6) | 0.86 (0.76–0.97)* | 1.03 (0.90–1.17) | 0.84 (0.74–0.95)** |
| Medicines for anxiety/ depression | 190 (4.7) | 187 (4.6) | 195 (4.8) | 1.04 (0.96–1.13) | 1.04 (0.98–1.11) | 1.00 (0.94–1.06) |
| Medicines for sleep problems | 189 (4.6) | 203 (5.0) | 207 (5.1) | 1.05 (0.94–1.17) | 1.00 (0.91–1.09) | 1.05 (0.96–1.15) |
| Mental health service use[2] | 341 (8.4) | 345 (8.5) | 317 (7.8) | 0.95 (0.83–1.09) | 0.92 (0.82–1.04) | 1.03 (0.91–1.16) |

[1]According cut-off score of ≤ 59. [2]Psychiatrist/psychologist/psychotherapist in past 12 months. aOR = Odds ratio adjusted for sex, age, marital status, employment status, education level, and disease at T1, T2, and T3. 95% CI = 95% confidence interval for aOR. All results are based on the weighted sample. Due to weighting, numbers may slightly differ between tables.

* $p < .05$,

** $p < .01$

For fatigue, no significant changes in the prevalence were found between T1, T2 and T3 across the four age categories. Only among respondents 50–64 years old, the prevalence of impaired functioning due to health problems differed significantly to some extent: the prevalence was lower at T3 (10.6%) compared with T1 (12.2; aOR = 0.78, 95% CI = 0.62–0.99, p = .039, but not compared with T2 (10.0%).

The use of medicine for anxiety and depression among respondents 18–34 years old was significantly more prevalent at T3 (3.4%) than at T2 (2.9%; aOR = 1.21, 95% CI = 1.01–1.46, p = .040) but not at T1. Concerning the use of medicines for sleep problems, no significant changes in the prevalence were found between T1, T2, and T3 across the four age categories.

Finally, the use of mental health professionals among respondents 65 years old and older decreased significantly in the 12 months before T3 (2.2%) compared with T1 (3.5%; aOR = 0.60, 95% CI = 0.41–0.87, p = .006) and T2 (3.1%; aOR = 0.69, 95% CI = 0.49–0.97, p = .031). No significant differences were found between T1 and T2 (for details see S1 Appendix Age categories).

## Incidence of mental health problems and service use of total study sample

Table 3 shows the incidence of mental health problems and service use at T2 and T3 among the total study sample and the results of crosswise multivariate logistic regression analyses (A1 incidence versus B2 incidence, and B1 incidence versus A2 incidence) showing that the incidence of mental health problems and service use at T2 and T3 did not differ significantly. Similar analyses using the cut-off ≤ 44 of the MHI-5, showed that the incidence of severe anxiety and depression symptom levels at T2 (3.7%) and T3 (3.3%) also did not significantly differ (results not shown in table).

## Incidence of mental health problems among age categories

Due to the cell counts of the incidence of mental health problems and service use at T2 and T3 (see Table 3) we limited the (cross-wise) multivariate logistic regression analyses among the four age categories to the incidence of anxiety and depression symptoms. The events-per-

**Table 3. Incidence of mental health problems and service use (N = 4,064).**

| | November-December 2018 (T1) | November-December 2019 (T2) | November-December 2020 (T3) | A1 versus B2 | B1 versus A2 |
|---|---|---|---|---|---|
| **Incidence[1]** | n (%) | n (%) | n (%) | aOR (95% CI) | aOR (95% CI) |
| Anxiety and depression symptoms[2] | n.a. | 293 (8.6) | 274 (8.1) | 1.00 (0.78–1.29) | 0.95 (0.75–1.22) |
| Sleep problems | n.a. | 120 (3.7) | 93 (2.9) | 0.86 (0.59–1.25) | 0.71 (0.47–1.07) |
| Fatigue | n.a. | 112 (4.0) | 93 (3.4) | 0.79 (0.53–1.19) | 0.87 (0.59–1.28) |
| Impaired functioning due to health problems | n.a. | 173 (4.7) | 205 (5.5) | 1.34 (1.00–1.80) | 1.06 (0.78–1.42) |
| Medicines for anxiety/depression | n.a. | 14 (0.4) | 20 (0.5) | 1.63 (0.56–4.77) | 1.31 (0.54–3.20) |
| Medicines for sleep problems | n.a. | 38 (1.0) | 36 (0.9) | 1.19 (0.64–2.21) | 0.76 (0.38–1.54) |
| Mental health service use | n.a. | 158 (4.2) | 136 (3.7) | 0.99 (0.72–1.38) | 0.77 (0.55–1.09) |

[1]Prevalence mental health problems among those without problems /use on previous survey.

[2]According to cut-off score of $\leq 59$ of the MHI-5.

[3]Psychiatrist/psychologist/psychotherapist in past 12 months. n.a. = not available because 2017 survey was outside aim of this study. aOR = Odds ratio adjusted for sex, age, marital status, employment status, education level and disease at T1 and T2, and T2 and T3. 95% CI = 95% confidence interval for aOR. A1 = incidence T2 of subgroup A. B1 = incidence T2 of subgroup B. A2 = incidence T3 of subgroup A. B2 = incidence T3 of subgroup B. All results are based on the weighted sample. Due to weighting numbers may slightly differ between tables.

variable (EPV) ratio for the other dependent variables, with seven predictors including control variables, became lower than 10. The results of these analyses showed no significant differences in incidence between T2 and T3 among 18–34 years old respondents ($T2^{incidence} = 12.4\%$, $T3^{incidence} = 13.1\%$), 35–49 years old respondents ($T2^{incidence} = 9.9\%$, $T3^{incidence} = 9.7\%$), 50–64 years old respondents ($T2^{incidence} = 6.2\%$, $T3^{incidence} = 5.0\%$), and 65 years and older respondents ($T2^{incidence} = 6.1\%$, $T3^{incidence} = 4.8\%$). Using the cut-off $\leq 44$ of the MHI-5 also did not reveal significant differences between the T2 and T3 incidence (results not shown in table).

## Risk factors of mental health problems and service use

In Table 4, the results of the multivariate logistic regression analyses are presented (for 95% confidence intervals of the adjusted OR's, see S2 Appendix Risk factors). Table 4 shows that existing mental health problems and service use (e.g., assessed one year earlier) were by far the strongest predictors for problems and use before and after the COVID-19 outbreak. For example, about 90% of the respondents with existing sleep problems and fatigue at T1 and T2, had sleep problems and fatigue at T2 and T3, respectively. With respect to anxiety and depression symptoms at T2 and T3, almost the same predictors were significant for these symptoms at T2 and T3: only those with a medium education level were no longer less at risk of symptoms than those with a relatively low education level. An almost similar pattern can be observed for sleep problems, only males and females did no longer differ in sleep problems after the outbreak (T3). However, 35–49 years old respondents were more at risk of sleep problems after the outbreak (at T3), but not before (at T2) relative to 65+ respondents. The patterns of risk factors of fatigue at T2 and T3 are identical. For impaired functioning due to health problems, e.g., that physical health or emotional problems hinder respondents' work over the past month, for instance, in their job, housekeeping, or in school, the results show that significant differences between subgroups disappeared after the outbreak: those with a medium and high education level compared with those with a relative low level, males compared with females, and youngest adult group (18–34 years) compared with 65+ respondents no longer differed in impaired functioning due to health problems. Finally, for mental health service use those employed before the outbreak (T2) used services significantly less often after the outbreak (T3)

**Table 4. Risk factors of mental health problems and service use (N = 4,064).**

| Predictors previous year | Anxiety and depression symptoms[1] 2019 (T2) n (%) | aOR | 2020 (T3) n (%) | aOR | Sleep problems 2019 (T2) n (%) | aOR | 2020 (T3) n (%) | aOR | Fatigue 2019 (T2) n (%) | aOR | 2020 (T3) n (%) | aOR |
|---|---|---|---|---|---|---|---|---|---|---|---|---|
| **Employed** | | | | | | | | | | | | |
| • no (ref.) | 378 (20.4) | 1 | 349 (19.1) | 1 | 499 (26.9) | 1 | 487 (26.7) | 1 | 660 (35.6) | 1 | 639 (35.0) | 1 |
| • yes | 309 (14.0) | 0.59*** | 337 (15.1) | 0.76* | 372 (16.8) | 0.86 | 382 (17.1) | 0.76 | 632 (28.6) | 0.76 | 628 (28.0) | 0.87 |
| **Married** | | | | | | | | | | | | |
| • yes (ref.) | 262 (13.4) | 1 | 251 (12.7) | 1 | 406 (20.7) | 1 | 398 (20.2) | 1 | 572 (29.2) | 1 | 550 (27.9) | 1 |
| • no | 425 (20.2) | 1.35** | 435 (20.8) | 1.27* | 466 (22.1) | 0.99 | 471 (22.5) | 1.14 | 720 (34.2) | 1.11 | 717 (34.3) | 1.26 |
| **Education level** | | | | | | | | | | | | |
| • low (ref.) | 217 (21.4) | 1 | 203 (20.5) | 1 | 286 (28.2) | 1 | 274 (27.6) | 1 | 337 (33.2) | 1 | 330 (33.3) | 1 |
| • medium | 247 (16.8) | 0.74* | 255 (17.7) | 0.79 | 311 (21.2) | 0.83 | 303 (21.0) | 0.97 | 499 (34.0) | 0.89 | 479 (33.2) | 0.97 |
| • high | 223 (14.1) | 0.76* | 228 (14.0) | 0.67** | 274 (17.3) | 0.83 | 292 (17.9) | 1.09 | 456 (28.8) | 0.88 | 457 (28.0) | 0.73 |
| **Sex** | | | | | | | | | | | | |
| • male (ref.) | 294 (14.7) | 1 | 319 (15.9) | 1 | 314 (15.7) | 1 | 316 (15.8) | 1 | 495 (24.7) | 1 | 495 (24.7) | 1 |
| • female | 393 (19.1) | 1.18 | 367 (17.8) | 0.93 | 557 (27.0) | 1.48** | 553 (26.8) | 1.29 | 797 (38.7) | 1.17 | 771 (37.4) | 1.07 |
| **Physical disease** | | | | | | | | | | | | |
| • no (ref.) | 503 (15.2) | 1 | 501 (15.3) | 1 | 582 (17.6) | 1 | 558 (17.1) | 1 | 901 (27.3) | 1 | 859 (26.3) | 1 |
| • yes | 184 (24.2) | 1.51** | 184 (23.1) | 1.76*** | 289 (38.0) | 1.66** | 310 (38.9) | 2.14*** | 391 (51.4) | 1.76** | 407 (51.0) | 1.93*** |
| **Age category** | | | | | | | | | | | | |
| • 65+ (ref.) | 117 (12.1) | 1 | 110 (10.5) | 1 | 237 (24.5) | 1 | 250 (23.8) | 1 | 283 (29.2) | 1 | 302 (28.7) | 1 |
| • 50–64 years | 159 (15.1) | 1.77*** | 148 (13.9) | 1.73* | 290 (27.5) | 1.79* | 291 (27.4) | 1.64* | 344 (32.7) | 1.18 | 346 (32.6) | 1.38 |
| • 35–49 years | 187 (19.5) | 2.58*** | 199 (21.3) | 3.25*** | 185 (19.3) | 1.30 | 186 (19.9) | 1.72* | 314 (32.7) | 1.17 | 296 (31.7) | 1.43 |
| • 18–34 years | 224 (20.7) | 2.26*** | 229 (22.5) | 3.19*** | 160 (14.8) | 1.35 | 142 (13.9) | 1.10 | 352 (32.5) | 1.50 | 323 (31.7) | 1.46 |
| **Mental health problems/service use** | | | | | | | | | | | | |
| • no (ref.) | 293 (8.6) | 1 | 274 (8.1) | 1 | 120 (3.7) | 1 | 93 (2.9) | 1 | 112 (4.0) | 1 | 93 (3.4) | 1 |
| • yes | 394 (59.8) | 13.36*** | 412 (60.0) | 14.31*** | 776 (89.1) | 214.07*** | 776 (89.1) | 243.4*** | 1174 (90.8) | 276.4*** | 1174 (90.8) | 263.68*** |

| Predictors previous year | Impaired functioning due to health problems 2019 (T1) n (%) | aOR | 2020 (T3) n (%) | aOR | Mental health service use[2] 2019 (T2) aOR | 2020 (T3) n (%) | aOR |
|---|---|---|---|---|---|---|---|
| **Employed** | | | | | | | |
| • no (Ref.) | 216 (11.7) | 1 | 214 (11.7) | 1 | 1 | 165 (9.0) | 1 |
| • yes | 129 (5.8) | 0.60** | 134 (6.0) | 0.54*** | 0.82 | 152 (6.8) | 0.46*** |
| **Married** | | | | | | | |
| • yes (ref.) | 160 (8.2) | 1 | 164 (8.3) | 1 | 1 | 119 (6.0) | 1 |
| • no | 185 (8.8) | 1.01 | 184 (8.8) | 1.05 | 1.02 | 198 (9.5) | 1.11 |
| **Education level** | | | | | | | |
| • low | 122 (12.0) | 1 | 104 (10.5) | 1 | 1 | 67 (6.8) | 1 |
| • medium | 116 (7.9) | 0.60** | 135 (9.4) | 1.16 | 0.81 | 122 (8.5) | 1.09 |
| • high | 107 (6.8) | 0.69* | 110 (6.7) | 0.99 | 0.98 | 128 (7.8) | 1.18 |

(*Continued*)

**Table 4.** (Continued)

| | | | | | | | | |
|---|---|---|---|---|---|---|---|---|
| **Sex** | | | | | | | | |
| • male (ref.) | 120 (6.0) | 1 | 140 (7.0) | 1 | 121 (6.0) | 1 | 106 (5.3) | 1 |
| • female | 225 (10.9) | 1.54** | 209 (10.1) | 1.14 | 224 (10.9) | 1.59** | 211 (10.2) | 1.62** |
| **Physical disease** | | | | | | | | |
| • no (ref.) | 204 (6.2) | 1 | 203 (6.2) | 1 | 262 (7.9) | 1 | 235 (7.2) | 1 |
| • yes | 141 (18.5) | 2.32*** | 145 (18.2) | 2.28** | 83 (10.9) | 1.52* | 82 (10.3) | 1.63** |
| **Age category** | | | | | | | | |
| • 65+ (ref.) | 74 (7.6) | 1 | 87 (8.3) | 1 | 30 (3.1) | 1 | 24 (2.3) | 1 |
| • 50–64 years | 105 (10.0) | 1.95** | 114 (10.8) | 1.95*** | 86 (8.2) | 2.91*** | 77 (7.3) | 4.13*** |
| • 35–49 years | 86 (9.0) | 2.51*** | 80 (8.6) | 2.02** | 100 (10.4) | 3.67*** | 104 (11.1) | 7.52*** |
| • 18–34 years | 80 (7.4) | 2.00** | 67 (6.6) | 1.40 | 129 (11.9) | 4.01*** | 112 (11.0) | 5.77*** |
| **Mental health problems/service use[2]** | | | | | | | | |
| • no (ref.) | 173 (4.7) | 1 | 136 (3.7) | 1 | 158 (4.2) | 1 | 136 (3.7) | 1 |
| • yes | 181 (52.5) | 10.99*** | 181 (52.5) | 8.78*** | 181 (52.5) | 22.53*** | 181 (52.5) | 21.98*** |

[1] According cut-off score of ≤ 59 on MHI-5.

[2] Psychiatrist/psychologist/psychotherapist in past 12 months. aOR = Odds ratio logistic regression analyses, adjusted for all other variables is table assessed at the same year. 95% CI = 95% confidence interval for aOR. ref. = Reference category. All results are based on the weighted sample. Due to weighting numbers may slightly differ between tables.

* $p < .05$,

** $p < .01$,

*** $p < .001$

than unemployed, in contrast to the period before the outbreak (T1-T2). Because of the relative low prevalence of medicines for anxiety/depression and medicines for sleep problems, we have omitted these dependent variables from the analyses.

## Discussion

The aims of the present prospective population-based study were to examine the *prevalence*, *incidence*, and *risk factors* for mental health problems and mental health service use among the adult general population 9 months after the COVID-19 outbreak, compared with before the outbreak (November-December 2018 and 2019). To optimize the representativeness of the study sample (N = 4,064), data were weighted using 16 demographic profiles of the adult Dutch population.

The main conclusion that can be drawn from the present study is that the prevalence, incidence, as well as risk factors of the assessed mental health problems and service use appear to be very stable among the general population (cf. [35]). We found no indications that on population level, the prevalence or incidence of anxiety and depression symptoms, sleep problems, fatigue, impaired functioning due to health problems, use of medicines for sleep problems, anxiety and depression, and mental health service use increased in November-December 2020, compared with pre-COVID-19 mental health and service use in November-December 2018 and in 2019. Results showed a small significant *decrease* in the prevalence of anxiety and depression symptoms at T3 (13.3%) compared with T1 (14.6%) and T2 (15.1%), and a small significant *decrease* in the mental health service use at T3 (2.2%) compare to T1 (3.5%) and T2 (3.1%). Based on the results of previous COVID-19 studies, we also examined the prevalence among 18–34 years, 35–49 years, 50–64 years, and 65 years and older respondents separately, showing almost identical results. Results did not show differences between mental health problems and mental health service use at T1 and T2 on the one side, and problems and use at T3 on the other side, except among 50–64 years old respondents and among 65+ old respondents. Our results about mental health problems seem to differ from almost all identified COVID-19 studies in the UK and US, but these studies focused on mental health problems until the summer of 2020.

Importantly, in our previous study [20], assessing anxiety and depression symptoms until the summer of 2020 (June), like the study of Breslau et al. [16] and Hyland et al. [36] no increase in prevalence compared with November-December 2019 was observed. Van Tilburg et al. [19] reported a similar finding among older respondents. As described, in the study of Breslau et al. [16] also no differences between the past-month prevalence of serious psychological distress in May 2020 and past-year prevalence assessed in February 2019 were found. In addition, they [16] found an incidence of 3.2% of serious distress according to K6. Although we used the MHI-5, this seems close to our incidence number for severe anxiety and depression symptom levels at T2 (3.7%) and T3 (3.3%). The extent to which the incidence of mental health problems after the COVID-19 outbreak in the UK and USA differed from the incidence before the outbreak is unknown. Nevertheless, in our previous study [18] we found no indications that the recovery of symptoms after the outbreak differed from the recovery before the outbreak. A recent report by Statistics Netherlands [46] showed that in the period July 2020-September 2020, 83% of the Dutch residents were positive about their general health, compared with 79–80% in the same period in 2017, 2018 and 2019.

In our study we focused on the mental health effects of this pandemic at the end of 2020, nine months after the outbreak. The identified studies with pre-COVID-19 outbreak data on mental health (see Introduction) were aimed at the mental health effects during the first months after the outbreak. This differences in study period may be of relevance. The meta-

analysis by Robinson et al. [37] of peer-reviewed and all other non-peer reviewed population-based studies (until January 11, 2011) with pre-COVID-19 data found that "*There was a small increase in mental health symptoms soon after the outbreak of the COVID-19 pandemic that decreased and was comparable to pre-pandemic levels by mid-2020 among most population sub-groups and symptom types*". In the recent post-outbreak USA population-based study by Riehm et al. [38] a similar pattern was found: "*by August 1, the odds of mental distress had returned to levels comparable to March 11.*"

We are not aware of prospective population-based COVID-19 studies to compare our findings on sleep problems, fatigue, use of pharmaceuticals, and mental health service use to. For example, sleep problems were assessed in many studies [39–41], but all studies used cross-sectional study designs and many were based on convenience samples.

How can we, given the outcomes of several studies showing an increase in mental health problems [9–17], understand the contradicting results of the present study [20, 42]?

Several elements, other than the resilience of people [3–5], may play a role and may help explain these differences, but future empirical research is required to examine and test these elements and explanations. First, unemployment might be a cause for increases in mental health problems [43] but unemployment remained relatively stable in the Netherlands during the COVID 19 outbreak, whereas they increased strongly in the USA in March and April 2020. However, also in the UK the unemployment figures did not rise as dramatically as in the USA in the first two months after the outbreak. From May 2020 on, due to the COVID-19 recovery programs put in place, the employment figures declined again strongly in the USA whereas they remained stable in the UK and the Netherlands [44]. Hence, the unemployment incidence might explain some of the differences with the USA, but they cannot explain the differences with the UK findings.

Analyses of the UK Household Longitudinal Study data showed an increase until the summer of 2020 in contrast to the Netherlands [20]. Another possible explanation could be that the Brexit process and the final end of the UK membership of the EU in 2020, similar to the USA due to the presidential elections, strongly divided the country, resulting in societal tensions and uncertainties about the future outside the EU. These tensions and uncertainties may have caused stress and therefore may have increased the risk of mental health problems [45–47]. With respect to the political climate, other than in the USA where the COVID-19 pandemic became part of an intense political debate and elections, to date this pandemic did not result in a similar political discourse in the Netherlands as in the USA.

Furthermore, mental disorders were much more prevalent in the USA than in the Netherlands, which may increase the risk for higher post-outbreak mental health problems [48]. It might be that the Dutch welfare and mental health care system provided better support to people with mental health issues during the COVID-19 outbreak or to people who experienced mental health issues after the outbreak than the USA or UK welfare system and mental health care system does. Unemployed adults (or adults who lost their jobs due to this pandemic) can invoke for unemployment benefits and, in principle, each Dutch citizen has health care insurance regardless of being employed or not in contrast to for example the USA.

Finally, we used other questionnaires to examine mental health than the UK and USA studies did (for instance, we used the MHI-5 and not the GHQ-12 as in the UK and K6 as in the USA studies). It is unknown if our study would have yielded other results, specifically a strong increase in mental health problems after the COVID-19 outbreak, when we had administered the GHQ-12 or K6 instead of the MHI-5. However, given the high correlations between such mental health measures [49] we do not consider that very likely.

## The role of pre-COVID mental health and stressors

The absence of an increase of mental health problems between T1 and T2 on the one side and T3 on the other side does not indicate that the COVID-19 outbreak has not negatively affected the mental health of individuals. As shown by the incidence rates, a minority suffers from mental health problems not experienced one year earlier that may be partly related to the disruptive effects of the COVID-19 pandemic. In addition, besides this pandemic, on a yearly basis, about 40% of adults were confronted with potentially traumatic and life-events that increase the risk for mental health problems among those confronted with these events [50, 51]. There are no valid reasons to assume that those events put mental health less at risk because of this COVID-19 pandemic. The stress sensitization model even suggests the opposite [52]. In other words, without ignoring the disruptive effects of this pandemic, we should be aware this pandemic does not occur in a vacuum. The very strong predictive values of pre-existing mental health problems clearly demonstrate, as in the study by Breslau et al. [16], this aspect. Perhaps this pandemic partly reveals mental health problems and patterns of mental health problems that were already present before the pandemic but, because of the large (media and scientific) attention towards the mental health effects [7, 53], have become more visible because of the pandemic. An increased visibility of mental health problems should not be confused with an (strong) increased prevalence of mental health problems. In either way, this underlines the relevance of nonretrospective pre-COVID mental health assessment among post-COVID-19 population-based probability study samples.

## Strengths and limitations

Although our results showed very clear patterns, some limitations need to be discussed when interpreting and using the outcomes of this study. We did not conduct clinical interviews that would certainly have enriched our study. Although our results do not point in the direction of a strong post-COVID-19 outbreak increase of mental disorders, future studies using clinical interviews will provide further insight into this topic. We extended previous prospective COVID-19 studies by assessing sleep problems, fatigue, use of medicines for sleep problems, anxiety and depression, impaired functioning due to health problems, and mental health service use. As described, sleep problems, fatigue and impaired functioning were assessed by one item questions. Future research is needed to examine the course of different aspects of sleep problems, fatigue and impaired functioning. In addition, future studies focusing on other mental health problems such as eating disorders, panic attacks, phobias, alcohol and drug misuse, and low self-esteem are warranted. The three surveys had a one-year time interval. Although we examined anxiety and depression symptoms in March 2019, March 2020, and June 2020 in previous studies, we cannot rule out the possibility that significant increases and decreases can be observed using shorter time intervals. The last survey in the present study was conducted in November-December 2020. It is unclear to what extent the results can be generalized to other developed countries who were hit harder by the COVID-19 pandemic. Finally, this study focused on adults and did not include children and adolescents who might be affected differently. At T3, the youngest respondent was 20 years old. Nevertheless, the prospective study design, the use of a large population-based probability sample, high response rates, that the non-response was not related to our dependent variables, and the weighting of data using 16 demographic profiles of the Dutch population are major strengths of the present study.

## Final conclusions

An important lesson that can be drawn from all prospective studies to date is that the large majority of the general population in general was capable, even though this pandemic

disrupted many aspects of life, to preserve their mental health during 2020. Similar to before the outbreak, a small part of the general population developed mental health problems during the pandemic that were not present before. Interestingly, the study by Pan et al. [54] showed that among people with depressive, anxiety, or obsessive-compulsive disorders, symptoms did not increase during the first months after the outbreak (cf. [20]). We offered possible explanations for the differences in results between the studies in the UK and USA on the one hand and the Netherlands on the other hand. Future comparative multi-country studies, including middle and low-income countries, are needed to further disentangle the complex relationships between especially existing welfare and health care systems, the size and dosage of governmental financial support programs, political systems and tensions, preventive measures, and mental health among the general population. Ongoing monitoring of the mental health of the general population is needed because the duration of this pandemic may undermine the capacity of individuals to cope with the consequences on the longer term.

## Supporting information

**S1 Appendix. Age categories.**
(DOCX)

**S2 Appendix. Risk factors.**
(DOCX)

## Author Contributions

**Conceptualization:** Peter G. van der Velden, Miquelle Marchand, Marcel Das, Ruud Muffels, Mark Bosmans.

**Data curation:** Peter G. van der Velden, Miquelle Marchand.

**Formal analysis:** Marcel Das.

**Investigation:** Peter G. van der Velden, Miquelle Marchand, Marcel Das, Ruud Muffels, Mark Bosmans.

**Methodology:** Peter G. van der Velden, Miquelle Marchand, Marcel Das, Ruud Muffels, Mark Bosmans.

**Supervision:** Peter G. van der Velden.

**Validation:** Peter G. van der Velden, Miquelle Marchand, Marcel Das, Ruud Muffels, Mark Bosmans.

**Visualization:** Peter G. van der Velden.

**Writing – original draft:** Peter G. van der Velden.

**Writing – review & editing:** Peter G. van der Velden, Miquelle Marchand, Marcel Das, Ruud Muffels, Mark Bosmans.

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
