## [Decision Letter · Decision Letter 0]

5 Jun 2021

PONE-D-21-04880

The prevalence, incidence and risk factors of mental health problems and mental health services use before and 9 months after the COVID-19 outbreak among the general Dutch population. A 3-wave prospective study.

PLOS ONE

Dear Dr. van der Velden,

Thank you for submitting your manuscript to PLOS ONE. After careful consideration, we feel that it has merit but does not fully meet PLOS ONE’s publication criteria as it currently stands. Therefore, we invite you to submit a revised version of the manuscript that addresses the points raised during the review process.

Both reviewers found the topic of the research to be important and timely. They also agreed that the use of a large population-based longitudinal data set was a clear strength of the study. However, both reviewers raised some concerns about the lack of theoretical clarity for the rationale of the research, and noted that the rationale and findings could be better positioned within relevant social, epidemiological, and political contexts. Doing so would allow for more meaningful comparisons between the Dutch population and other relevant populations. These and other thoughtful comments from the reviewers should be given full and careful consideration in your revision.

We look forward to receiving your revised manuscript.

Kind regards,

Fuschia M. Sirois, PhD

Academic Editor

PLOS ONE

Journal Requirements:

Reviewers' comments:

Reviewer's Responses to Questions

**Comments to the Author**

1. Is the manuscript technically sound, and do the data support the conclusions?

Reviewer #1: Partly

Reviewer #2: Yes

2. Has the statistical analysis been performed appropriately and rigorously? 

Reviewer #1: Yes

Reviewer #2: Yes

3. Have the authors made all data underlying the findings in their manuscript fully available?

Reviewer #1: Yes

Reviewer #2: Yes

4. Is the manuscript presented in an intelligible fashion and written in standard English?

Reviewer #1: Yes

Reviewer #2: Yes

5. Review Comments to the Author

Reviewer #1: I read this paper a few times, because as a reader I felt its presentation was confusing and I wanted to make sure I had not missed the point. For example, the introduction has sections delineated as ‘results’ which come before the methods section. I was unsure what these ‘results’ meant and why they came before the methods section. The research design was also unclear with different designs alluded to ranging from repeated measures in the abstract to prospective in the introduction, if it was a repeated measures prospective study it would have been clearer to use the same terminology throughout for consistency. There is clarity in the well-performed statistical analysis, but the rationale for the focus on mental health, sleep and medication use was unclear.

I found the statement confusing about the lack of studies on sleep and COVID -19, because there are numerous studies on sleep and the relationship with COVID-19 including some longitudinal and some systematic reviews. Given how multi-factorial mental health is and the unique situation presented by the global pandemic, I wondered about the utility and how the interpretation of results could be used to positive effect within the field of mental health. I was also unsure as to the validity of an accurate comparison with the USA and UK when the measures differed between all the studies mentioned. Furthermore, in The Netherlands some questions failed to use validated measures, merely singular questions that were ‘similar’ to questions taken from validated measures.

Within the discussion, specifically pages 30-31, rhetorical reasoning appears, for example, suggesting Brexit as a reason for higher UK scores, or societal tensions dividing the UK and USA, or even COVID-19 and the surrounding political discourse. Unfortunately, there is no scientific evidence presented for these claims, which I felt weakened the paper. Furthermore, suggesting support in health systems as a potential reason for differences in results is problematic when services are structured and funded differently. This again fails to address the complexity. Considering these claims overall left the discussion feeling somewhat unfocused, as if unsure of the story it wanted to tell. On arriving at the end of the paper, as a reader I wondered about originality and addition to the evidence base because of the previous papers using the same data set from the LISS panel and suggesting similar findings.

Reviewer #2: This study examined the effects of COVID-19 pandemic on a comprehensive set of well-being outcomes including anxiety and depressive symptoms, sleep problems, fatigue, disabilities due to health problems, use of medicine for anxiety and depression, and mental health. Data were from the Longitudinal Internet studies for the Social Sciences (LISS) (N=4,064). This analysis is an extension to earlier covid related work using LISS investigating a longer time period and additional well-being outcome. Somewhat surprisingly, the authors found not significant increase in any of these outcomes. The authors conclude Dutch adults are resilient.

While the study was well done, there are still some issues that need to be addressed:

1. It would be useful to provide more depth on the social, political, and epidemiological context in the Netherlands to provide more of a global perspective. Some of this is mention in the Discussion, but elaboration in the Introduction to "set the stage" would be most useful.

- For example, how wide spread was testing? Mitigation efforts? Government response? Public opinion?

2. The authors speculate a bit for *why* there were no differences, yet other comparative countries, on balance, report lower well-being during pandemic. The reasons provided are not very well developed and need more fleshing out. This is a limitation in the paper.

3. Finally, what can the international community learn from the Dutch experience?

4. Another major limitation is there is not theoretical development in the Introduction of the paper. The authors briefly mention "conservation of resources" ... but how does that operate? Provided a theoretical framework will help with rationale for analytic approach, interpretation of results and providing additional substantive meaning.

A few other, more minor, and editorial issues:

a. Typo on pg 16. The authors mention NS for non-response, yet writ "p<.05"

b. It would also be useful to know N with all 3 T, just 2, just T1 and T3 etc.

c. Internet surveys have notorious low response rates. How was the sample chosen? What makes it representative of the Dutch population? These issues weren't spelled out in the Methods.

d. The first paragraph of the Discussion simply repeats the aims; this needs reworking.

6. PLOS authors have the option to publish the peer review history of their article (what does this mean?). If published, this will include your full peer review and any attached files.

Reviewer #1: No

Reviewer #2: No

---

## [Author Response · Author response to Decision Letter 0]

15 Jun 2021

Fuschia M. Sirois, PhD

Academic Editor

PLOS ONE

Dear Dr. Sirois,

Thank you very much for reviewing our manuscript and the opportunity to revise our manuscript according to the clear and helpful comments of the two reviewers. 

We believe that we were able to address all 18 comments of the reviewers and that the comments helped us to improve our manuscript. We tried to better clarify the rationale and contexts of our study as much as possible by revising specific paragraphs and by adding new paragraphs. 

Below we have described in detail how we responded to each comment. We have marked the revisions in yellow in the revised manuscript and we hope that this revised manuscript meets your criteria.

Kind regards, also on behalf of my co-authors,

Peter G. van der Velden

Corresponding author

REVIEWER 1

1. I read this paper a few times, because as a reader I felt its presentation was confusing and I wanted to make sure I had not missed the point. For example, the introduction has sections delineated as ‘results’ which come before the methods section. I was unsure what these ‘results’ meant and why they came before the methods section.

Response:

We regret that the headings caused confusion. The three “results” sections in the introduction were about the results of published prospective studies in different countries and not about the results of our present study.

Following this comment, we therefore added a new subheading to better clarify for the reader that these sections provide an overview of the results of previous prospective studies on the mental health of the general population:

Results of previous prospective COVID-19 studies on the mental health of the general population (corps 16).

Being aware that (almost) accepted but not published peer-reviewed studies are missed, to date we identified 12 studies (written in English) among the general population in the United Kingdom (UK), United States of America (USA) and the Netherlands that were based on probability samples of the general population with non-retrospective data on pre-COVID-19 mental health. Below we first provide a summary of the main outcomes of these studies. Given the aim of the present study (see below), we focus on the prevalence of mental health problems among the total study samples and different age categories.

Based on this comment we also revised the subheading of the sections devoted to the results in different countries (was corps 16, now corps 14):

Results of prospective studies in the UK 

Results of prospective studies in the USA 

Results of prospective studies in the Netherlands 

2. The research design was also unclear with different designs alluded to ranging from repeated measures in the abstract to prospective in the introduction, if it was a repeated measures prospective study it would have been clearer to use the same terminology throughout for consistency. 

Response

We fully agree with the reviewer that consistency in terms is very important. Based on this comment we have checked our manuscript to certify that we use the same terms throughout the manuscript. We used the description “Longitudinal Internet studies for the Social Sciences (LISS) panel” for the present study. Although we agree with the need for consistency, we cannot of course change the name of this panel and replace longitudinal with prospective.

The terms “prospective” and “repeated measures”, as for example used in the abstract, refer to two different aspects, e.g. the study design and statistical analyses and are therefore not inconsistent. Our study has “a prospective study design”. The analyses we conducted are “repeated measures multivariate logistic regression analyses”, but perhaps the reviewers meant something else.

3. There is clarity in the well-performed statistical analysis, but the rationale for the focus on mental health, sleep and medication use was unclear.

Response:

We are glad to notice that the reviewer considers our statistical analyses clear and well-performed. 

We have revised the section on the Conservation of Resources theory to clarify better the rationale for our study. We would also like to refer to our response to comment 14. We believe that this revision, in combination with the other paragraphs of the introduction including the overview of population based studies on mental health problems before and after this pandemic, provides a clear rationale for the focus on mental health (including sleep and use).

4. I found the statement confusing about the lack of studies on sleep and COVID -19, because there are numerous studies on sleep and the relationship with COVID-19 including some longitudinal and some systematic reviews. 

Response:

When submitting our manuscript, none of the identified prospective studies among the general population with pre-COVID data assessed differences in sleep problems before and after the COVID-19 outbreak. We are aware of the study with pre-COVID data on sleep by Evans (2021), but this study was conducted among UK university undergraduates (of a specific university) and not the general population (they found no pre- post-outbreak differences). A recent search did not identify a new COVID peer-reviewed study among the general population on sleep with pre-outbreak data on sleep after we submitted our manuscript in February 2021.

We assume the reviewer refers to the systematic reviews on sleep by Souza et al. (2021), Jahrami et al. (2021) and Nochaiwong et al. (2021). These reviews were published after of just before we submitted our manuscript and therefore could not be included in the original manuscript. Importantly, the inclusion criteria and tables of the three reviews (with included studies published in the first months after the outbreak) show that all studies were cross-sectional in nature and often based on convenience samples. We hope that the reviewer agrees with our argument that without pre-COVID data on sleep (or comparable pre-COVID reference groups), no conclusions can be made about the effects of this pandemic on sleep (such as change in prevalence and incidence).

Based on this comment we added in the discussion (added text in Italics)

We are not aware of prospective population based COVID-19 studies to compare our findings on sleep problems, fatigue, use of pharmaceuticals, and use of mental health services use with. For example, sleep problems were assessed in many studies [38-40], but all studies used cross-sectional study designs and many were based on convenience samples. 

5. Given how multi-factorial mental health is and the unique situation presented by the global pandemic, I wondered about the utility and how the interpretation of results could be used to positive effect within the field of mental health. 

Response:

Thank you for this interesting question. We believe that reviewer 2 asked a somewhat similar question. We have tried to answer the question of this reviewer and reviewer 2 simultaneously. Therefore, we would like to refer to our response to comment 13.

6. I was also unsure as to the validity of an accurate comparison with the USA and UK when the measures differed between all the studies mentioned. Furthermore, in The Netherlands some questions failed to use validated measures, merely singular questions that were ‘similar’ to questions taken from validated measures.

Response:

We agree with the reviewer that we did not mention and discuss the possibility that differences in measures could be the reason for differences in outcomes between the UK and US studies on the one hand, and the Dutch studies on the other hand. We therefore added (added text in italics):

“Furthermore, mental disorders in the US were much more prevalent in the US than in the Netherlands which may increase the risk for higher post-outbreak mental health problems [40]. It might be that the Dutch welfare system provide better support to people with mental health issues during the COVID-19 outbreak or to people who ran into mental health issues after the outbreak than the US or UK welfare system does.

 Finally, we used other questionnaires to examine mental health than the UK and USA studies did (for instance, we used the MHI-5 and not the GHQ-12 as in the UK and K6 as in the USA studies). It is unknown if our study would have yielded other results, specifically a strong increase in mental health problems after the COVID-19 outbreak, when we had administered the GHQ-12 or K6 instead of the MHI-5. However, given the high correlations between such mental health measures [51] we do not consider that very likely.”

7. Within the discussion, specifically pages 30-31, rhetorical reasoning appears, for example, suggesting Brexit as a reason for higher UK scores, or societal tensions dividing the UK and USA, or even COVID-19 and the surrounding political discourse. Unfortunately, there is no scientific evidence presented for these claims, which I felt weakened the paper. 

Response: 

In the section the reviewer refers to we tried to discuss differences in outcomes: what are possible reasons for these differences in outcomes (such as differences in unemployment, period in which studio took place, the prevalence of mental disorders before the outbreak, the health and welfare systems between countries). 

We disagree with the reviewer’s comment that we made any claim with respect to the role of Brexit and presidential election in the US. We only offered a possible explanation. We wrote “Another possible explanation could be that the Brexit process and final end of the UK membership of the EU in 2020, similar to the US due to the presidential elections, strongly divided the country resulting in societal tensions and uncertainties about the future outside the EU. These tensions and uncertainties may have increased the risk of mental health problems. With respect to the political climate, other than the US where the COVID-19 pandemic became part of an intense political debate and elections, to date this pandemic did not result in a similar political discourse in the Netherlands as in the US.”. 

However, we agree with the reviewer that we should have added references to support offering this possible explanation. We therefore added the following three publications in this section (we revised the sentence slightly)

“These tensions and uncertainties may cause stress and therefore may have increased the risk of mental health problems [47-49]”.

[47] APA. Presidential Election a Source of Significant Stress for More Americans than 2016 Presidential Race. https://www.apa.org/news/press/releases/2020/10/election-stress (accessed June 7 2021).

[48] O'Neill S. Brexit and Northern Ireland: leaders must consider the mental health of the population. Lancet Psychiatry. 2019; 6: 372-373. https://doi.org/10.1016/S2215-0366(19)30121-X

[49] Smith KB, Hibbing MV, Hibbing JR. Friends, relatives, sanity, and health: The costs of politics. PLoS ONE 2019; 14: e0221870. https://doi.org/10.1371/journal.pone.0221870

8. Furthermore, suggesting support in health systems as a potential reason for differences in results is problematic when services are structured and funded differently. This again fails to address the complexity. 

Response:

Based on this comment we revised this section into: 

“Furthermore, mental disorders in the US were much more prevalent in the US than in the Netherlands which may increase the risk for higher post-outbreak mental health problems [50]. It might be that the Dutch welfare and mental health care system provide better support to people with mental health issues during the COVID-19 outbreak or to people who ran into mental health issues after the outbreak than the US or UK welfare system and mental health care system does. Unemployed adults (or adults who lost their job due to this pandemic) can invoke for unemployment benefits and, in principle, each Dutch citizen has a health care insurance regardless of being employed or not in contrast to for example the US.”

9. Considering these claims overall left the discussion feeling somewhat unfocused, as if unsure of the story it wanted to tell. On arriving at the end of the paper, as a reader I wondered about originality and addition to the evidence base because of the previous papers using the same data set from the LISS panel and suggesting similar findings.

Response:

Following the comments of the reviewer we have revised some sections of the discussion and believe that we were able to address the comments as much as possible. We fully realize that our paper is not able to answer all relevant questions and that future research is needed to address these questions.

In our previous papers we also used the LISS panel but certainly not the same data. The current manuscript used for the first time data on mental health of the November-December 2020 survey. In addition, we examined for the first time prevalence, incidence and risk factors for sleep problems, fatigue, disabilities due to health, medicines for anxiety/depression, medicines for sleep problems, and use of mental health services among the general population in 2018, 2019 and 2020.

 

REVIEWER 2

10. This study examined the effects of COVID-19 pandemic on a comprehensive set of well-being outcomes including anxiety and depressive symptoms, sleep problems, fatigue, disabilities due to health problems, use of medicine for anxiety and depression, and mental health. Data were from the Longitudinal Internet studies for the Social Sciences (LISS) (N=4,064). This analysis is an extension to earlier covid related work using LISS investigating a longer time period and additional well-being outcome. Somewhat surprisingly, the authors found not significant increase in any of these outcomes. The authors conclude Dutch adults are resilient.

Response:

We agree with this summary. With respect to reviewer’s surprise, we would also like to refer to our response to comment 12.

11. While the study was well done, there are still some issues that need to be addressed:

1. It would be useful to provide more depth on the social, political, and epidemiological context in the Netherlands to provide more of a global perspective. Some of this is mention in the Discussion, but elaboration in the Introduction to "set the stage" would be most useful.

- For example, how wide spread was testing? Mitigation efforts? Government response? Public opinion?

Response:

Thank you for your kind compliment.

Based on this comment we added at the end of introduction (added text in Italics).

With respect to COVID-19 in the Netherlands, soon after the outbreak the Dutch government implemented large financial support programs for companies who significantly lost revenues because of the COVID-19 pandemic to allow them to keep people employed. In addition, governmental taxes were postponed. The Dutch governmental deficit increased in 2020 with 40 billion euro to 435 billion euro largely due to these financial programs [26]. In 2020 about 800.000 residents were tested positive for COVID-19, although the number of residents with COVID-19 in 2020 is presumably higher [27]. After the outbreak residents and companies were confronted with several (partial) lockdowns in 2020, including closed schools and universities. Statistics Netherlands (CBS) reported that about 169,000 had died in the Netherlands, 10% more than expected compared to previous years [28].”

We left the public opinions out of this summary because its takes a lot of space to address this topic correctly, and public opinions are outside the aim of the present study. 

12. The authors speculate a bit for *why* there were no differences, yet other comparative countries, on balance, report lower well-being during pandemic. The reasons provided are not very well developed and need more fleshing out. This is a limitation in the paper.

Response:

We are not sure what the reviewer is asking for because we offered several possible explanations (based on the comments of reviewer 1, see comment 6, we have added an alternative reason, e.g. that we used other research instruments), such as low unemployment rates in the Netherlands, the Dutch governmental financial support programs for companies and organizations, the Dutch health care and welfare system where everybody has a health insurance, the absence of political tensions about COVID-19 as in the US, the absence of Brexit-related stress, and differences between the US and Netherlands in the prevalence of mental disorders (before this pandemic). Perhaps the reviewer want to share which other possible explanations the reviewer is thinking of.

However, after we re-read the discussion section with this comment in mind, we noticed that we did not clarify enough that the identified UK and US studies assessed mental health until the summer of 2020, while our studies examined the mental health at the end of 2020. This difference in study period may be relevant because of a published (on medRxiv) but not yet peer-reviewed review and meta-analysis of peer-reviewed and many other non-peer reviewed COVID-19 studies with pre-COVID-19 data on mental health (this study was published on a preprint server after we submitted our manuscript) by Robison et al. (2021: see https://doi.org/10.1101/2021.03.04.21252921). We became aware of this study because the authors requested some additional information about our study.

They found that “The overall increase in mental health symptoms was most pronounced during the early stages of the pandemic (March-April), before decreasing and being generally comparable to pre-pandemic levels by mid-2020…. Findings confirm that the initial outbreak of the pandemic was associated with a significant but statistically small increase in mental health symptoms” (extracted from abstract).

Following the comment of the reviewer and using this recent meta-analyses of true prospective studies, we added in the discussion section: 

“In our study we focused on the mental health effects of this pandemic at the end of 2020, nine months after the outbreak. The identified studies with pre-COVID-19 outbreak data on mental health (see introduction) were aimed at the mental health effects during the first months after the outbreak. This differences in study period may be of relevance. A published but not yet peer-reviewed meta-analysis by Robinson et al. [37] of peer-reviewed and all other non-peer reviewed population-based studies (until January 11, 2011) with pre-COVID-19 data found that “The overall increase in mental health symptoms was most pronounced during the early stages of the pandemic (March-April), before decreasing and being generally comparable to pre-pandemic levels by mid-2020…. Findings confirm that the initial outbreak of the pandemic was associated with a significant but statistically small increase in mental health symptoms”. In the recent post-outbreak US population-based study by Riehm et al. [38] a similar pattern was found: by August 1, the odds of mental distress had returned to levels comparable to March 11.” 

13. Finally, what can the international community learn from the Dutch experience?

Response:

Reviewer 1 had a somewhat similar question (see comment 5). Below we try to answer both questions simultaneously.

We believe it is a little bit too early to describe what can be learned from the Dutch experience (for both the international community and the field of mental health), although we tried to explain why our results differ from studies in the US and UK. Our possible explanations suggest that financial support programs, existing adequate social welfare and access to health care, absence of dominant political tensions and politization of preventive measure during the pandemic, etc., probably mitigate the negative effects of this pandemic. 

Based on both comments we added at the end of the discussion: 

“Final remarks

In either way, an important lesson that can be drawn from all prospective studies to date is that the large majority of the general population was capable to preserve their mental health during 2020, although this pandemic disrupted many aspects of life. Like in the years before this pandemic, a small part of the general population developed mental health problems that were not present before (in this case) the COVID-19 pandemic. Interestingly, the study by Pan et al. [56] showed that among people with depressive, anxiety, or obsessive-compulsive disorders symptoms did not increase during the first months after the outbreak (cf. [20]). We offered possible explanations for the differences in results between the studies in the UK and US on the one hand and the Netherlands on the other hand. Future comparative multi-country studies, including middle and low income countries, are needed to further disentangle the complex relationships between especially existing welfare and health care systems, the size and dosage of governmental financial support programs, political systems and tensions, preventive measures, and the mental health among the general population.” 

14. Another major limitation is there is not theoretical development in the Introduction of the paper. The authors briefly mention "conservation of resources" ... but how does that operate? Provided a theoretical framework will help with rationale for analytic approach, interpretation of results and providing additional substantive meaning.

Response:

Based on this comment we added (added text in italics)

“Based on the Conservation of Resources (COR) theory of Hobfoll [3,4] we may expect that this ongoing pandemic has negative effects among the general population since it directly or indirectly threatens important resources such as safety and health (for instance by being infected, loss of a significant other), social contacts and support (for instance by social distancing and staying-at-home following lockdowns), work and income (for instance by loss of job or reduced work) among the general population. Resource loss is an important factor, like existing problems, in predicting the impact of stressful events on mental health such as this pandemic. However, according to the COR model [3,4] people also strive to obtain, retain, protect resources and to restore lost resources (such as social contacts, employment, housing, health) and their resilience should not be underestimated [3-6]. The duration of this pandemic may however undermine the capacities of individuals, communities, and countries to cope with the negative effects of this pandemic on the medium and longer term. This may cause stress and increase the risk for mental health problems.” 

15. A few other, more minor, and editorial issues:

Typo on pg 16. The authors mention NS for non-response, yet writ "p<.05"

Response:

Thank you for this comment. We have replaced “….the non-response was not significantly (p < 0.05) associated with….” with “….the non-response was not significantly (p > 0.05

) associated with….”. 

16. It would also be useful to know N with all 3 T, just 2, just T1 and T3 etc

Response:

Based on this comment we first added in the abstract.

“In total, 4,064 respondents with complete data participated at all three surveys.”

It is a little effort to provide these numbers, but we hesitate to include the numbers of those participated at T1 and T2 only (N=558), T1 and T3 only (N=190) , and T2 and T3 only (N=272).

In the method section we wrote “The total study sample consisted of 4,064 respondents with complete data across the three surveys (99%)”.

17. Internet surveys have notorious low response rates. How was the sample chosen? What makes it representative of the Dutch population? These issues weren't spelled out in the Methods.

Response:

We are very sorry, but we do not fully understand this comment. In the method section we described how the sample was chosen “This panel (LISS) is based on a traditional probability sample drawn from the Dutch population register of 16 years and older by Statistics Netherlands and administered by Centerdata”. 

In addition, because not all respondents at T1 participated at T2 and T3 “We next weighted the data using 16 exclusive demographic profiles among the total adult Dutch population to optimize the representativeness of the current study, based on the data of Statistics Netherlands (see: https://opendata.cbs.nl/#/CBS/en/; in English). The 16 profiles were constructed using the variables sex (male, female), age (18-34, 35-49, 50-64, 65 years and older) and marital status (married and unmarried), yielding 2*4*2=16 demographic profiles. All findings are based on the weighted sample”. We therefore consider our sample representative of the Dutch population (of 16 years and older).

Regarding the response we clarified that “Data on mental health of adults were extracted from the longitudinal Health module, in particular the waves in November 2018 (T1: Ninvited =6,466, response=84.4%), in November 2019 (T2: Ninvited=5,954, response=86.4%), and in November 2020 (T3: Ninvited=6,832, response=83.6%). These response rates cannot be considered low.

Perhaps the reviewer got the impression that respondents themselves took the initiative to sign up for the LISS panel and participate. However, in the LISS panel respondents cannot sign up for the panel: they were asked to participate after Statistics Netherlands selected them (based on a probability sample of the Dutch population):

We therefore added (added text in italics). 

“This panel is based on a traditional probability sample drawn from the Dutch population register of 16 years and older by Statistics Netherlands and administered by CentERdata. People cannot sign up themselves as a respondents for the LISS panel, so there is no issue of self-selection bias.” 

18. The first paragraph of the Discussion simply repeats the aims; this needs reworking.

Response:

We agree with the reviewer that this section was too long. We have shorted this section into. 

“The aims of the present prospective population-based study were to examine the prevalence, incidence, and risk factors for mental health problems and mental health services use among the adult general population 9 months after the COVID-19 outbreak, compared to before the outbreak (November-December 2018 and 2019). To optimize the representativeness of the study sample (N=4,064), data were weighted using 16 demographic profiles of the adult Dutch population.”

---

## [Decision Letter · Decision Letter 1]

9 Aug 2022

PONE-D-21-04880R1The prevalence, incidence and risk factors of mental health problems and mental health services use before and 9 months after the COVID-19 outbreak among the general Dutch population. A 3-wave prospective study.PLOS ONE

Dear Dr. van der Velden,

Thank you for submitting your manuscript to PLOS ONE. After careful consideration, we feel that it has merit but does not fully meet PLOS ONE’s publication criteria as it currently stands. Therefore, we invite you to submit a revised version of the manuscript that addresses the points raised during the review process.

ACADEMIC EDITOR:The text has improved with revisions. However, because it is unconventionally structured, our readers may interpret it more as a research report than an original article. My suggestion is to reorder the topics (introduction, methods, results and discussion) and subtopics as suggested by the STROBE checklist. My suggestion for the topics and subtopics:Introduction: no subtopics: indicate what the study is about, why it was carried out, why it should be published, a brief summary of previous evidence, what is unknown about the research topic and the objectives.Methods: with the subtopics: design, scenario, sample selection, data collection, statistical methods and ethical aspects.Results: with the subtopics: sample composition, main finding, secondary findings.Discussion: with the subtopics: synthesis of research results, validity, comparison with the literature (move here the part of previous studies present in the introduction), interpretation of findings (move the interpretations present in the results here), conclusions.Please ensure that your decision is justified on PLOS ONE’s publication criteria and not, for example, on novelty or perceived impact.

We look forward to receiving your revised manuscript.

Kind regards,

Marcus Tolentino Silva

Academic Editor

PLOS ONE

Reviewers' comments:

Reviewer's Responses to Questions

**Comments to the Author**

1. If the authors have adequately addressed your comments raised in a previous round of review and you feel that this manuscript is now acceptable for publication, you may indicate that here to bypass the “Comments to the Author” section, enter your conflict of interest statement in the “Confidential to Editor” section, and submit your "Accept" recommendation.

Reviewer #1: All comments have been addressed

Reviewer #3: (No Response)

2. Is the manuscript technically sound, and do the data support the conclusions?

Reviewer #1: Yes

Reviewer #3: Partly

3. Has the statistical analysis been performed appropriately and rigorously? 

Reviewer #1: Yes

Reviewer #3: Yes

4. Have the authors made all data underlying the findings in their manuscript fully available?

Reviewer #1: Yes

Reviewer #3: (No Response)

5. Is the manuscript presented in an intelligible fashion and written in standard English?

Reviewer #1: Yes

Reviewer #3: Yes

6. Review Comments to the Author

Reviewer #1: The authors have done well to address comments so speedily and thoroughly. All queries have been answered satisfactorily. The paper now explains reasoning and substantiates the suggestions within the paper.

Reviewer #3: - page 14: there is a typo, the range of the Cronbach alpha is 0 - 1, not 100.

- measures need more description: are they validated tools? do they measure a construct or are they a checklist? if they measure a construct, what is the intarnal structure?

- why were not calculated the cronbach alpha for all the measures?

- Acronyms:

- acronyms should be written in long form in the original language and then in their english form.

- I do not see the extended form of the MOS acronym.

- What does the acronym 'GEE' in the data analysis section means?

- acronym aOR is repeated in its long form after declared in the short one

- consider that p values cannot exceed 1, the authors may consider to report them without the zero before the decimal dot (APA-7 style).

cross-wise meaning?

- Table 4, 1: CIs of ORs need to be reported to understand the significance of results.

- Table 4, 2: reading and intepreting this table is not straightforward, it is possible to use another format/way of presenting it?

- in general, results are very detailed and I think that graphical representations of results are needed to give the readers information with an instant glance.

7. PLOS authors have the option to publish the peer review history of their article (what does this mean?). If published, this will include your full peer review and any attached files.

Reviewer #1: No

Reviewer #3: No

---

## [Author Response · Author response to Decision Letter 1]

22 Aug 2022

We gratefully thank reviewer 1 and 3, and the editor for the thoughtful and helpful comments. It helped us to further improve our manuscript. Below we have described how we responded to each comment. 

COMMENTS EDITOR. 

1) The text has improved with revisions. However, because it is unconventionally structured, our readers may interpret it more as a research report than an original article. My suggestion is to reorder the topics (introduction, methods, results and discussion) and subtopics as suggested by the STROBE checklist. My suggestion for the topics and subtopics:

Response:

Thank you for your suggestions. We have revised the manuscript using several suggestions you offered, to prevent a possible confusion as much as possible.

 However, we do not fully understand your comment that our manuscript was unconventionally structured (besides the intro, which we revised (see below)) because we followed the PLOSONE guidelines (https://journals.plos.org/plosone/s/submission-guidelines) and the structure and sub headings hardly differs from our earlier papers published in PLOSONE (or that of other papers in PLOSONE covering social science research). For instance, PLOSONE requires a section “Materials and methods” while you (like many other journals) suggest the word “Methods”. Because in our experience the office is critical on this issue, we chose to use their wordings. 

1. Introduction: no subtopics: indicate what the study is about, why it was carried out, why it should be published, a brief summary of previous evidence, what is unknown about the research topic and the objectives.

Response:

Following this comment we have eliminated the subheadings with respect to results of studies conducted in the USA, UK and Netherlands (which we added earlier based on a comment of a previous reviewer). To further improve the readability of the introduction (prevent a lengthy intro without sub headings) we also shortened the introduction by around 35%. 

2. Methods: with the subtopics: design, scenario, sample selection, data collection, statistical methods and ethical aspects.

Response:

Please see our response to comment 1. We have replaced “Data analyses” with Statistical methods”. 

3. Results: with the subtopics: sample composition, main finding, secondary findings.

Response:

Please see our response to comment 1. We do not have secondary findings.

4. Discussion: with the subtopics: synthesis of research results, validity, comparison with the literature (move here the part of previous studies present in the introduction), interpretation of findings (move the interpretations present in the results here), conclusions. 

Response:

As described in our response to the comments we have revised/shortened the introduction section to a large extent. Because of this substantial revision we chose not to move parts of the intro to the discussion, also because we believe that the intro must show what is (not) known (at the time the research is conducted of course) and why the study is conducted. 

The editor furthermore suggested to move the interpretations present in the results in the discussion. We have checked the text on these issues but could not find clear examples of interpretations other than presentations/explanations of tables/findings. Perhaps the editor can give us examples because we agree that interpretations/comparisons with other studies, conclusions, etc. belong in the discussion section (we need the help of the editor on this issue).

Other revisions

The meta-analysis of Robinson et al. was published in 2022 in JAD. We have updated the references and text about this important study. We furthermore corrected small errors and revised unclear sentences.

REVIEWER #1: 

5. The authors have done well to address comments so speedily and thoroughly. All queries have been answered satisfactorily. The paper now explains reasoning and substantiates the suggestions within the paper.

Response

Thank you very much for your kind words of appraisal.

 

REVIEWER #3

6. page 14: there is a typo, the range of the Cronbach alpha is 0 - 1, not 100.

Response

Based on this comment we have replaced 0.85 with .85 .

7. Measures need more description: are they validated tools? do they measure a construct or are they a checklist? if they measure a construct, what is the internal structure?

Response

The MHI-5 is a validated and widely used inventory assessing anxiety and depression symptom levels. In research, as we did, the sum scores of the 5 item are used (see Ware et al., 1992; Means Christensen et al., 2005). The questions/single items (yes/no) such as sleep problems and fatigue are administered each year since the start of LISS panel in 2007 (and used in several other peer-reviewed studies (examples see below, and other papers we are preparing). Because all data of the LISS panel is open access, we assume that other researchers have used these variables too.

van der Velden PG, Bosmans MWG, van der Meulen E, Vermunt JK. Pre-event trajectories of mental health and health-related disabilities, and post-event traumatic stress symptoms and health: A 7-wave population-based study. Psychiatry Res. 2016;246:466-473. https://doi.org/10.1016/j.psychres.2016.10.024

van der Velden PG, Das M, Muffels R. The stability and latent profiles of mental health problems among Dutch young adults in the past decade: A comparison of three cohorts from a national sample. Psychiatry Res. 2019;282:112622. https://doi.org/10.1016/j.psychres.2019.112622

van der Velden PG, van Bakel HJA, Das M. Mental health problems among Dutch adolescents of the general population before and 9 months after the COVID-19 outbreak: A longitudinal cohort study. Psychiatry Res. 2022;311:114528. https://doi.org/10.1016/j.psychres.2022

8. Why were not calculated the cronbach alpha for all the measures?

Response

We computed the CA’s for the MHI-5 total scores. It is not possible to compute CA’s of single items, such as for “Do you regularly suffer from fatigue” and “How often did you use the following health services over the past 12 months?”

9. Acronyms should be written in long form in the original language and then in their english form.

Response

We have checked the manuscript and used the long format before using the acronyms (see also response next comment. 

10. I do not see the extended form of the MOS acronym.

Response

Following this comment we have revised:

“5-item sub scale of the MOS 36-item short-form health survey [30, 31]).

into:

“5-item sub scale of the Medical Outcomes Study (MOS), 36-Item Short Form Survey Instrument (SF-36, [30, 31])).

11. What does the acronym 'GEE' in the data analysis section means?

Response

Based on the comment we have revised the sentence into (revision in italics):

“To examine the extent to which the prevalence of the seven assessed mental health problems after the COVID-19 outbreak in November-December 2020 (T3) changed compared to the prevalence in November-December 2018 (T1) and 2019 (T2), generalized estimating equations (GEE) for longitudinal ordinal data were conducted (GENLIN in SPSS version 28, using an autoregressive working correlation structure) with problems at T1, T2 and T3 as dependent variables (separate analyses for each dependent variable).”

12. Acronym aOR is repeated in its long form after declared in the short one

Response

The acronym aOR and CI were clarified in the first paragraph “non-response” and under each table presenting aOR’s and CI.

13. Consider that p values cannot exceed 1, the authors may consider to report them without the zero before the decimal dot (APA-7 style).

Response

Based on this comment we have deleted all leading zero’s.

14. cross-wise meaning?

Response

We would like to refer to the section ”Statistical methods”(in previous version entitled “Data analyses”, in which we explained why and how we conducted the analyses. Because of this (lengthy) clarification we originally had chosen not to repeat this in the results section. However, based on this comment we revised the sentence:

“Table 3 shows the incidence of mental health problems and services use at T2 and T3 among the total study sample and the results of crosswise multivariate logistic regression analyses showing that the incidence of the mental health problems and services use at T2 and T3 did not differ significantly.”

Into:

“Table 3 shows the incidence of mental health problems and services use at T2 and T3 among the total study sample and the results of crosswise multivariate logistic regression analyses (A1 incidence versus B2 incidence, and B1 incidence versus A2 incidence) showing that the incidence of the mental health problems and services use at T2 and T3 did not differ significantly.”

15. Table 4, 1: CIs of ORs need to be reported to understand the significance of results.

- Table 4, 2: reading and interpreting this table is not straightforward, it is possible to use another format/way of presenting it?

Response

1) It seems that something went wrong. The CIs of the aORs were all presented in the S2 Appendix risk factors (this appendix was referred to in the beginning of paragraph “Risk factors of mental health problems and services use”). When submitting the current revised version we will check if the appendix are included in the PDF’s.

2) We understand the comment of the reviewer, but we looked several times at this issue but did not find a workable alternative for presenting the results of the analyses (splitting the table 4 in 5 tables did not solve the problem of the space needed to present the results). For the reason mentioned by the reviewer, we already replaced the CI’s to the appendix to improve the readability as much as possible (cf. Qian & Tahara, 2020, Table 3 (https://doi.org/10.1371/journal.pone.0235883). We are sorry, but we could not find a better way to present the results.

16. in general, results are very detailed and I think that graphical representations of results are needed to give the readers information with an instant glance.

Response

We thank the reviewer for the invitation to consider/think about a graphical presentation of our findings (besides/instead of a table(s)). We started the exercise with Table 4 and looked at the possibilities/usefulness of a forest-like graph in which the aORs and CIs are presented/plotted. We first noticed that this would require at least 5 graphs (because of the five dependent variables, with two rows for each predictor (2018, 2019). We next looked at the aORs in Table 4 to check if all aORs can be depicted in one forest plot for each dependent variable. Unfortunately, we had to conclude that this was not possible/workable because for instance the aORs for sleep problems and fatigue were rather high (≥ 214.07, not surprising given the high prevalence of sleep problems and fatigue among those with exiting sleep problems and fatigue ) compared to the aORs of for instance marital status (≤1.35). Including both aOR in one forrest plot would make the lower aORs almost invisible.

---

## [Decision Letter · Decision Letter 2]

14 Sep 2022

PONE-D-21-04880R2The prevalence, incidence and risk factors of mental health problems and mental health services use before and 9 months after the COVID-19 outbreak among the general Dutch population. A 3-wave prospective study.PLOS ONE

Dear Dr. van der Velden,

Thank you for submitting your manuscript to PLOS ONE. After careful consideration, we feel that it has merit but does not fully meet PLOS ONE’s publication criteria as it currently stands. Therefore, we invite you to submit a revised version of the manuscript that addresses the points raised during the review process.

ACADEMIC EDITOR: Please make the corrections marked by the reviewer.

We look forward to receiving your revised manuscript.

Kind regards,

Marcus Tolentino Silva

Academic Editor

PLOS ONE

Journal Requirements:

Reviewers' comments:

Reviewer's Responses to Questions

**Comments to the Author**

1. If the authors have adequately addressed your comments raised in a previous round of review and you feel that this manuscript is now acceptable for publication, you may indicate that here to bypass the “Comments to the Author” section, enter your conflict of interest statement in the “Confidential to Editor” section, and submit your "Accept" recommendation.

Reviewer #1: (No Response)

Reviewer #4: All comments have been addressed

2. Is the manuscript technically sound, and do the data support the conclusions?

Reviewer #1: Partly

Reviewer #4: Yes

3. Has the statistical analysis been performed appropriately and rigorously? 

Reviewer #1: Yes

Reviewer #4: I Don't Know

4. Have the authors made all data underlying the findings in their manuscript fully available?

Reviewer #1: Yes

Reviewer #4: Yes

5. Is the manuscript presented in an intelligible fashion and written in standard English?

Reviewer #1: No

Reviewer #4: Yes

6. Review Comments to the Author

Reviewer #1: This paper is improved from my initial reading and the decision to rewrite the introduction means it now makes much more sense. However, there remain some flaws in the work, which could be addressed.

Place the last paragraph on pp. 8-9 before the aim of the study so it flows neatly.

Ethical approval/informed consent p. 10; it would probably be wise to give the reason why ethics was not needed; aggregated anonymised data. Agree with mentioning that participants had agreed for further analysis of their data.

p.11, p. 16, Table 2, Table 3 etc.,: ‘disabilities due to mental health problems’. This could perhaps be worded as reduced or impaired functioning and use the ICF language. Disability using the social model refers to the barriers within society, used otherwise it places the ‘problem’ within the individual and makes the paper biomedical and lacking in consideration for social barriers. This makes people the problem. Means-Christensen et al. do not use the term disability in relation to psychological distress, they use impaired functioning (p. 566) which is the language of the ICF, as such the article is somewhat misrepresented.

Likewise, reword mental health problems throughout the paper to mental health difficulties. Language is extremely important so we do not reify and marginalise groups.

The term ‘eating problems’ could be amended. I thought these were eating disorders, are these identified by type in the LISS data? Then sleep problems are not identified, this could be a critique of the LISS data collection, does it mean disturbance in circadian rhythms, insomnia, or does it mean sleep disorders such as breathing-related sleep disorders? Alternatively, are a number of conditions/disorders aggregated under one heading? A little more clarity and critique here about the data would open the study out.

p.22: 2nd sentence, should it not be that existing mental health difficulties and service use were the strongest predictor for continued mental health difficulties and service use before and after the COVID-19 pandemic?

Services use should be service use throughout the paper, or use of services. Appreciate this a language issue.

p.29, 2nd paragraph: ‘people who ran into mental health issues’ people experience mental health difficulties, they do not ‘run into them’ please amend.

Limitations: this section goes further than strengths and limitations, so place some of this in a conclusion.

Final remarks section: I am not convinced of the rationale in the comments in this section because I feel it adds little to the paper. Offering differences between the UK, Netherlands and USA (observed from other studies, not analysed statistically) was not the aim of the paper and adding these last six lines detracts from the work done. The conclusion needs to focus on the paper, summing it up and its addition/contribution to existing research. I would delete this section and rewrite. Perhaps call it conclusion rather than final remarks begin with prospective studies and use some of the limitation section, which is more suited to a conclusion section. For example, the results suggesting further research using clinical interviews to provide more insight and future studies on mental health difficulties, eating disorders etc.

Reviewer #4: Dear authors,

I consider the comments raised by the reviewers in the previous round of reviews were successfully addressed. Congratulations for your effort on that.

7. PLOS authors have the option to publish the peer review history of their article (what does this mean?). If published, this will include your full peer review and any attached files.

Reviewer #1: No

Reviewer #4: No

---

## [Author Response · Author response to Decision Letter 2]

27 Sep 2022

We thank the reviewers very much for their time and effort. It certainly helped us to improve our manuscript.

Reviewer #1: 

1. This paper is improved from my initial reading and the decision to rewrite the introduction means it now makes much more sense. However, there remain some flaws in the work, which could be addressed.

Response:

Thank you for your kind words.

2. Place the last paragraph on pp. 8-9 before the aim of the study so it flows neatly.

Response:

We have moved this section as suggested and renumbered the references.

3. Ethical approval/informed consent p. 10; it would probably be wise to give the reason why ethics was not needed; aggregated anonymised data. Agree with mentioning that participants had agreed for further analysis of their data.

Response:

Based on this comment, we have revised the first sentence as follows:

“Since our research did not impose certain (experimental) behavior, our research did not need the approval of a Dutch Medical Ethical Testing committee according to the Dutch Law (see https://english.ccmo.nl/investigators/legal-framework-for-medical-scientific-research/your-research-is-it-subject-to-the-wmo-or-not).”

4. p.11, p. 16, Table 2, Table 3 etc.,: ‘disabilities due to mental health problems’. This could perhaps be worded as reduced or impaired functioning and use the ICF language. Disability using the social model refers to the barriers within society, used otherwise it places the ‘problem’ within the individual and makes the paper biomedical and lacking in consideration for social barriers. This makes people the problem. Means-Christensen et al. do not use the term disability in relation to psychological distress, they use impaired functioning (p. 566) which is the language of the ICF, as such the article is somewhat misrepresented.

Response:

Based on this comment and to prevent possible confusion we have replaced “disabilities” with “impaired functioning” throughout the manuscript, including the tables and appendix 1 and 2. 

5. Likewise, reword mental health problems throughout the paper to mental health difficulties. Language is extremely important so we do not reify and marginalise groups.

Response:

We fully agree with the reviewer that language is very important. However, we do not understand that the term “mental health problems” marginalizes groups. In addition, a search on PUBMED with the term “mental health problems” identified 16,461 papers while as search with the term "mental health difficulties" identified “only” 1,226 papers, clearly suggesting that “mental health problems” is an often used and well accepted used term (we used this term in all our papers). For these reasons we did not reword the term “mental health problems”. 

6. The term ‘eating problems’ could be amended. I thought these were eating disorders, are these identified by type in the LISS data? Then sleep problems are not identified, this could be a critique of the LISS data collection, does it mean disturbance in circadian rhythms, insomnia, or does it mean sleep disorders such as breathing-related sleep disorders? Alternatively, are a number of conditions/disorders aggregated under one heading? A little more clarity and critique here about the data would open the study out.

Response:

Based on this comment we have replaced “eating problems” with “eating disorders”. In addition, we added:

“As described, sleep problems, fatigue and impaired functioning were assessed by one item questions. Future research is needed to examine the course of different aspects of sleep problems, fatigue and impaired functioning”. 

7. p.22: 2nd sentence, should it not be that existing mental health difficulties and service use were the strongest predictor for continued mental health difficulties and service use before and after the COVID-19 pandemic?

Response:

We understand the comment of the reviewer, but to be able to state that existing mental health difficulties and service use were the strongest predictor for continued mental health difficulties and service use before and after the COVID-19 pandemic, different analyses are required. In this case the dependent variables should be persistent mental health problems and service use. 

8. Services use should be service use throughout the paper, or use of services. Appreciate this a language issue.

Response:

We have revised the text using the term “service use”.

9. p.29, 2nd paragraph: ‘people who ran into mental health issues’ people experience mental health difficulties, they do not ‘run into them’ please amend.

Response:

Thank you for noticing this error. We have corrected this mistake.

10. Limitations: this section goes further than strengths and limitations, so place some of this in a conclusion.

Response:

Based on this comment we moved the sentence “Monitoring of the mental health of the general population is needed because the duration of this pandemic on the longer term may undermine the capacity of individuals to cope with the consequences” to the final conclusions section.

11. Final remarks section: I am not convinced of the rationale in the comments in this section because I feel it adds little to the paper. Offering differences between the UK, Netherlands and USA (observed from other studies, not analysed statistically) was not the aim of the paper and adding these last six lines detracts from the work done. The conclusion needs to focus on the paper, summing it up and its addition/contribution to existing research. I would delete this section and rewrite. Perhaps call it conclusion rather than final remarks begin with prospective studies and use some of the limitation section, which is more suited to a conclusion section. For example, the results suggesting further research using clinical interviews to provide more insight and future studies on mental health difficulties, eating disorders etc.

Response:

Based on this comment we replaced “final remarks” with “final conclusions”.

We are sorry, but we disagree with reviewer’s comments on this section. It is the task of researchers to compare the findings of other researchers with their findings. This is what we did in the discussion section. We therefore do not understand reviewers remark “Offering differences between the UK, Netherlands and USA (observed from other studies, not analysed statistically) was not the aim of the paper ……”. In addition, in line with the comments of the reviewer we stated “Future comparative multi-country studies, including middle and low-income countries, are needed to further disentangle the complex relationships between especially existing welfare and health care systems, the size and dosage of governmental financial support programs, political systems and tensions, preventive measures, and mental health among the general population”.  

Reviewer #4: 

Dear authors,

12. I consider the comments raised by the reviewers in the previous round of reviews were successfully addressed. Congratulations for your effort on that.

Response:

Thank you for your compliments.

---

## [Decision Letter · Decision Letter 3]

17 Oct 2022

The prevalence, incidence, and risk factors of mental health problems and mental health service use before and 9 months after the COVID-19 outbreak among the general Dutch population. A 3-wave prospective study.

PONE-D-21-04880R3

Dear Dr. van der Velden,

We’re pleased to inform you that your manuscript has been judged scientifically suitable for publication and will be formally accepted for publication once it meets all outstanding technical requirements.

Kind regards,

Marcus Tolentino Silva

Academic Editor

PLOS ONE
---

## [Editor Report · Acceptance letter]

20 Oct 2022

PONE-D-21-04880R3 

The prevalence, incidence, and risk factors of mental health problems and mental health service use before and 9 months after the COVID-19 outbreak among the general Dutch population. A 3-wave prospective study. 

Dear Dr. van der Velden:

I'm pleased to inform you that your manuscript has been deemed suitable for publication in PLOS ONE. Congratulations! Your manuscript is now with our production department. 

Kind regards, 

on behalf of

Dr. Marcus Tolentino Silva 

Academic Editor

PLOS ONE